# Evaporation Rate of Colloidal Droplets of Jet Fuel and Carbon-Based Nanoparticles: Effect of Thermal Conductivity

**DOI:** 10.3390/nano9091297

**Published:** 2019-09-11

**Authors:** Ahmed Aboalhamayie, Luigi Festa, Mohsen Ghamari

**Affiliations:** Department of Mechanical Engineering and Engineering Management, Wilkes University, Wilkes-Barre, PA 18766, USA; ahmed.aboalhamayie@wilkes.edu (A.A.); luigi.festa@wilkes.edu (L.F.)

**Keywords:** droplet evaporation, nanofuel, carbon-based nanomaterials, thermal conductivity

## Abstract

Adding nanoparticles to liquid fuel is known to promote its combustion characteristics through improving several thermo-physical properties. This study investigates the effects of adding carbon nanoparticles on thermal conductivity and evaporation rate of liquid jet fuel. Multi-walled carbon nanotubes, activated carbon nanoparticles, and graphene nanoplatelets were added to jet fuel at different concentrations to prepare colloidal suspensions. Thermal conductivity is determined by passing known amounts of heat through a very thin layer of fuel and measuring temperature difference across its thickness. A fiber-supported droplet technique is also used to evaluate evaporation rate due to force convection of a hot inert gas. It is observed that both thermal conductivity and evaporation rate increase as a result of nanoparticle addition. Since there is no radiation heat transfer mechanism, the increase in evaporation rate is concluded to be only due to enhanced thermal conductivity.

## 1. Introduction

Recent advances in nanoparticle production has led to the emergence of a new type of heat transfer fluid known as nanofluid. Nanofluids, which are typically categorized as colloidal suspension of 1–100 nm nanoparticles in fluids, have been extensively reported to have superior heat transfer properties, such as higher specific heat and, in particular, higher thermal conductivity [1,2,3,4,5,6]. This, in turn, has led to growing interest and research in using nanofluids in a wide range of potential heat transfer applications such as in heat exchangers [7], micro-channel cooling blankets [8], high temperature drilling [9], light emitting diodes (LEDs) [10], etc.

Application of nanofluids has also led to a large body of research with the aim of developing nanofluid-type fuels with enhanced burning characteristics. In this new type of fuel, also known as nanofuel, nano-sized energetic materials are added to traditional liquid fuel. The energy added by particles as well nanofuel higher thermal conductivity results in enhanced ignition and combustion properties [11,12,13,14,15,16]. Several studies have also suggested that nanofuels have enhanced optical properties [14,17,18]. This becomes particularly important as thermal radiation plays a crucial role in liquid fuel combustion.

To promote combustion characteristics in liquid fuels, both metallic and non-metallic nanoparticles have been studied. Due to high reactivity, and yet less expensive production cost, aluminum particles have been mixed with different liquid fuels and examined over wide range of operating conditions [11,12,13,19,20,21]. Other metallic nanoparticles, such as aluminum-lithium alloys [22], boron [23,24], and iron [24], have also been tried and reported as satisfactory fuel additives. Nonetheless, emission of toxic metallic oxides as a combustion byproduct has set some limitations on the application of metallic nanoparticles in nanofuels. As a result, much attention has been drawn to carbon-based nanomaterials such as graphene/graphite [16,17,25], carbon nanotubes [16,26], activated carbon nanoparticles [16], and acetylene-black [15]. Indeed, it is the distinguished thermal properties of carbon nanomaterials that has made them an extraordinary candidate in developing nanofuels. For instance, by using molecular dynamics simulations, Berber et al. determined that the thermal conductivity of carbon nanotube can be as high as 6000 W/mK [27]. Graphene nanoplatelets have also been reported to have an in-plane thermal conductivity of 3000 W/mK [28]. Yet, there is not a strong consensus over the main mechanism leading to enhanced thermal conductivity in nanofluids. Some studies have attributed the high thermal conductivity to the nanoconvection as an indirect outcome of the Brownian motion of nanoparticles [29,30]. However, there are other studies that suggest the interfacial layer formed around nanoparticles acts as a thermal bridge between nanoparticles and improves heat transfer within nanofluids [1,31]. There is also another theory, suggesting that the formation of aggregates and their morphology can enhance thermal conductivity [32].

Adding carbon nanostructures to liquid fuel has also been reported to reduce fuel optical transmittance [18]. This is believed to be the major mechanism behind faster burning rate as fuels with lower transmittance can absorb more of the flame radiation and use it toward a faster evaporation, and hence, combustion. There are other studies, too, that have focused on quantifying the effect of improved optical properties in nanofuels [17,33]. The one issue present in experimental works examining optical properties is the strong coupling between radiation and conduction heat transfer. In other words, improved optical properties can help nanofuel to absorb more energy simply by transmitting less. Nonetheless, molecular diffusion and internal circulation can still significantly contribute to heat transfer processes within a droplet. To have a better understanding of each of these mechanisms, it is required to examine each effect independently. For this purpose, an experiment was developed to measure the evaporation rate of isolated droplets of nanofuel due to only force convection (i.e., no radiation source). The colloidal nanofuels were also tested for their thermal conductivity in order to understand the correlation between the improved thermal conductivity and the changes in fuel evaporation rate.

## 2. Materials and Methods

To examine the effect of thermal conductivity on the evaporation rate of nanofuels, both of these properties were measured independently and in two separate experiments. In conducting any successful measurement on a nanofuel, preparing a stable colloidal suspension of nanoparticle in fuel is a key step. While each droplet evaporation experiment usually takes less than 30 s, measuring conductivity may take up to 30 min, and hence, it is necessary to prepare suspensions that will remain stable throughout the whole duration of measurement. Mechanical stirring, sonication, and adding surfactants are common techniques to prevent or slow down particle agglomeration. Combining these methods has been reported to prolong the stability of nanofluids [14,17,18,34]. As a result, a hybrid method employing surfactant, agitation, and sonication was specifically developed to prepare stable nanofuels for this study.

In order to reduce the level of uncertainty in interpreting result and make more decisive conclusions, the same type of nanoparticles and base fuel used in one of the authors’ previous works [16,35] was considered for this study: activated carbon nanoparticles (CNP), multi-walled nanotubes (MWNT), and graphene nanoplates (GNP). A summary of the physical properties of particles are listed in Table 1. The difference in carbon contents is simply because of impurity present in nanoparticles, especially in CNP. It should be mentioned that the size and purity of nanoparticle can significantly affect the optical properties of nanofuels [18,36,37]. However, since there is no major chemical reaction present in this work, the impurity does not seem to have a significant impact on the parameters studied here. The phenomenon of interest in this study is the phase change of base fuel from liquid to vapor in which the solid nanoparticles do not actively participate. On the other hand, the dimension and morphology of nanoparticles can affect the way they promote energy transfer mechanisms through nanofuel. For instance, particles with larger specific surface area (SSA) are expected to be more efficient in terms of transferring heat.

In order to prepare stable suspensions of particles in base fuel, the following procedure adopted: initially, a 3% (by weight) solution of Sorbitan Monooleate, commercially known as Span 80, (C_24_H_44_O_6_, Sigma-Aldrich, Product Number 85548, St. Louis, MO, USA) in jet fuel (Jet A, supplied by a municipal airport) was prepared on a magnet stirrer for 10 min. Span 80 is a surfactant with a hydrophilic-lipophilic balance of 4.3 ± 1.0, and hence, easily dissolves in base fuel. While still stirring, nanoparticles were added to the solution at different weight concentrations and were allowed to mix for another 10 min until a visually homogeneous mixture with no particle lump was obtained. To improve stability of nanofuel and prevent quick agglomeration of nanoparticles, fuel samples were sonicated using an ultrasonic homogenizer (Omni International Sonic Ruptor 4000, Kennesaw, GA, USA). Only a 20 mL of each sample was prepared, and due to this small volume a 5/32” microtip at a power setting of 30% was used to convert ultrasonic energy into mechanical energy and transfer it to the samples. To achieve an optimum nanofuel preparation technique, various sonication powers as well as sonication times were tried until it was determined that intermittent sonication (0.5 s long pulses, apart by 0.5 s) for 30 min can yield nanofuels that will not go through phase separation for at least one hour after the sonication. The black color of carbon nanoparticles used in this study results in suspensions that completely block light right after sonication. However, as the particles coagulate, agglomerates are formed and begin to settle at the bottom of the sample container by the effect of gravity. As a result, carbon-in-fuel suspensions begin to turn murky at the liquid surface and gradually downward. Due to their morphology being cylindrical and also large specific surface area, carbon nanotubes are more susceptible to entangle and form agglomerates. This makes 3% MWNT in jet fuel the least stable nanofuel prepared in this work, which began to destabilize about 1 h after sonication.

## 3. Experimental Methods

### 3.1. Thermal Conductivity of Nanofuels

To measure thermal conductivity of nanofuel samples, a Hilton H470 unit was used. Figure 1 provides a schematic view of this device in which a thin radial layer of nanofuel sample (Δx=0.325 mm) was heated by a heating element with a known resistance R. The heating element can be powered by a transformer to generate a constant heat rate of $V^{2}/R,$ where V is the voltage adjusted by the transformer. A constant flow of 3 lit/min cold tap water was used in the cooling jacket around the fuel sample to dissipate the excess heat into the atmosphere and help achieve equilibrium. Two Type K thermocouples were also used to measure the temperature difference ΔT across the thin nanofuel layer. Thermal conductivity (k) of nanofuel sample can be then evaluated using Fourie’s law of heat conduction, $\dot{Q}=-kA\Delta T/\Delta x$, where A is the cylindrical surface area of the fuel sample subject to the heat rate $\dot{Q}$.

### 3.2. Evaporation Rate of Nanofuel Droplet

The experimental method developed in this part of the study consisted of a fiber-supported droplet arrangement along with a flow of a hot inert gas, as shown in Figure 2. Small droplets with diameters of ~1-mm or less were generated by a micro-syringe and placed on the intersection of three silicon carbide fibers (NL-202 Nicalon™) [38]. Nitrogen at a flow of 27 L/min and heated to 120 °C by an in-house heating system was used as an inert medium to evaporate fuel droplets. The custom designed heating system is 170 cm of ¼ inch OD copper tubing, formed into a of 15 cm diameter coil. A 182 cm, 310 W insulated heating tape (BriskHeat BIH051060L, Columbus, OH, USA) wrapped around the copper coil provided a constant flux of heat to the nitrogen gas running through the tube. The heated nitrogen was then directed into a 10 cm long, 12 mm ID upright aluminum manifold and, from there, exited through a honeycomb mesh in order to produce a uniform convective flow. Two rectangular aluminum plates attached to the plungers of two pull solenoids were mounted above the manifold outlet and under the droplet to let the nitrogen flow toward the droplet only when the gas temperature reached the pre-set value of 120 °C. To reduce heat loss, copper coil and aluminum manifold were covered by glass wool and foam rubber pipe insulations, respectively. The temperature of nitrogen delivered toward droplet was measured by a type K thermocouple, placed 1 inch below the droplet where the gas leaves the manifold through the honeycomb. A temperature controller (BriskHeat SDC120KF-A, Columbus, OH, USA) was then used to power off the heating tape when the temperature reached the pre-set value.

Due to very small diameter of SiC fibers (~16 μm), the surface tension is just large enough to let the droplet remain attached to the fibers and still keep its spherical shape during evaporation process. To keep track of droplet size during heating process, backlit images of droplet were recorded using a high-speed camera (IDT XS-1440p) at a rate of 50 frames per second. Images were processed in NASA Spotlight [39,40] to remove the fibers from the field of view and determine the instantaneous droplet diameter, D, in pixels. A 1/32” diameter bearing ball was later used to calibrate the field of view and determine the droplet initial diameter, $D_{0}$, in millimeters. Droplet evaporation rate constant, $K_{evp}$, was then determined as the slope of the linear equation fitted to the data on a ${\left(D/D_{0}\right)}^{2}$ versus $t/D_{0}^{2}$ diagram as described by the *d^2^*-law of evaporation:(1)$${\left(\frac{D}{D_{0}}\right)}^{2}=1-K_{evp}\left(\frac{t}{D_{0}^{2}}\right)$$

## 4. Results and Discussion

### 4.1. Thermal Conductivity of Nanofuels

Nanoparticles were introduced into a mixture of surfactant and jet fuel at weight concentrations of up to 3% and using the protocol explained in Section 2. Higher concentrations were prepared and found to become unstable quickly and sometimes before the conductivity measurement ends. That is why 3% was considered as the highest concentration for this part of the work. Three different heat rates were supplied to each sample and the temperatures across the liquid layer were monitored until equilibrium was established. For the three heating cases, the variation of fuel mean temperature (defined as the arithmetic mean of equilibrium temperatures measured by the two thermocouples) was found to be very insignificant (± 4 °C). This, in turn, resulted in small variation of thermal conductivity at each particle concentration as shown in Figure 3.

The data presented in Figure 3 clearly show significant improvement in thermal conductivity as a result of adding nanoparticles. In general, all three types of nanoparticles bring some enhancement in thermal conductivity of nanofuel. Higher increase was observed at higher particle concentration, except for CNP where thermal conductivity begins to drop at concentrations above 2%. It is worth mentioning that fuel preparation as well as measurements were performed by another operator and at all CNP concentrations, too. Yet, the same pattern in variation of thermal conductivity was observed. Between all three tested nanoparticles, a maximum increase of 29% in thermal conductivity was obtained at 3% MWNT concentration. This is believed to be due to extraordinarily high thermal conductivity of carbon nanotubes which has been measured to approach that of natural diamond at 3000 W/mK [41] or even exceed it and reach 6600 W/mK [27].

### 4.2. Evaporation Rate of Nanofuels

The evaporation rates were determined experimentally for the baseline jet fuel and varying concentration of different nanoparticles. For this purpose, CNP, MWNT, and GNP were suspended in a mixture of surfactant and jet fuel at weight fractions of up to 2%. Higher concentrations would clog the microsyringe needle, making it difficult to generate droplets, and therefore were not considered in this work. At each concentration, droplet evaporation experiment was conducted for a minimum of 7 droplets, and evaporation rate constants were extracted from diagrams similar to those shown in Figure 4.

From the results, all particle types conform to the *d^2^*-law of evaporation (equation 1) for the initial 65% of fuel evaporation (by volume) corresponding to ${\left(D/D_{0}\right)}^{2}$ of nearly 0.5. In general, all colloidal droplets go through similar stages, as shown in Figure 4, regardless of nanoparticle type or concentrations: Stage A of the figure (${\left(D/D_{0}\right)}^{2}>1$) exhibits the swelling behavior of the droplet at the initial time of the process. The evaporation in this zone is characterized by an instantaneous diameter greater than that of the initial diameter, as caused by thermal expansion of the base fuel. Stage B ($0.5<{\left(D/D_{0}\right)}^{2}<1$) depicts the stable suspension zone of the evaporation process. This zone is characterized by a linear decrease in square dimeter with respect to time in accordance with the *d^2^*–law of evaporation. The evaporation rates as summarized in this experiment were extracted in this zone and are represented by the slope of the curve.

Stage C (${\left(D/D_{0}\right)}^{2}<0.5$) depicts the unstable suspension zone of the evaporation curve. This zone is characterized by variable evaporation rates, due to increased particle concentration (as a result of base fuel evaporation) and formation of larger particle agglomerates. All of the diagrams displayed in this stage of Figure 4 show a nonlinear behavior in such a way that the tangent to the data, also known as instantaneous evaporation rate, reduces by time. The graphical representation of a proposed model explaining this reduction has been displayed by Figure 5a. In this model, and due to preferential evaporation, the more volatile species (i.e., base fuel) will evaporate first. This results in an increase in the local population of particles, leading to higher probability of agglomerate formation. Once these agglomerates form larger clusters, they can trap the base fuel and suppress its transport to the droplet surface where the phase change from liquid to fuel vapor occurs. The snapshots displayed by Figure 5b clearly show the formation of big clusters as evaporations evolves for a 0.25% MWNT in jet fuel droplet.

The quantitative data used to determine the evaporation characteristics include evaporation rates and percent increase over the baseline fuel. The average evaporation rates as a function of concentration are exhibited in Figure 6 for all three nanoparticles used in this study. It is seen from this figure that the introduction of CNP into jet fuel directly increases the evaporation rate by 51%. Enhanced thermal conductivity can be considered as the main reason for such a substantial increase in evaporation rate as conduction is the main heat transfer mechanism within the droplet. For MWNT samples, the evaporation rates vary with concentration, but in general more than 33% increase was observed for all of the concentrations with a maximum percent increase as high as 58.8%. The decrease in evaporation rates for concentrations at 2% are assumed to be contributed to the resistance particles apply to base fuel species as they transport to the droplet surface. The evaporation rate percent increase for GNP samples was also found to be typically above 35% except for the 1.5 and 2% concentration and possibly due to higher resistance as previously discussed.

GNP suspensions exhibit the lowest enhancement of evaporation characteristics which can be in part due to its lower thermal conductivity at the same concentration among all three nanoparticles used in this study. GNP particles have a planar morphology with a thermal conductivity of 3000 W/mK in the direction parallel to surface but only 6 W/mK perpendicular to surface [28]. While large particle width (5 μm) and high thermal conductivity across the width could be advantageous in terms of transferring heat in planar direction and within each particle, transfer of heat to and from the base fuel mainly occurs normal to the planar surface. Having the lowest specific surface area among three nanoparticle types as well as very low thermal conductivity perpendicular to the surface results in lower thermal conductivity in GNP colloidal suspensions which can itself explain lower evaporation rates.

The empirical values obtained from the measurements and analysis suggest enhanced evaporation rate for colloidal suspensions over the baseline jet fuel. A comparison of the evaporation results is provided and offered in Table 2. The experimental results offered in this table show the significance of introducing nanoparticles into fuel. The data suggests that the highest evaporation rate can be obtained between particle concentrations of 1.0% and 1.5%. It is believed that faster agglomeration of nanoparticles and higher resistance against the base fuel flow toward the droplet surface are responsible for the decrease in evaporation rates at 2% and greater concentrations.

## 5. Conclusions

The effect of adding three types of carbon-based nanoparticles on thermal conductivity and evaporation rate of jet fuel droplets was investigated in this experimental work. In general, adding small amounts of nanoparticles can promote thermal conductivity with the highest increase of 29% obtained at 3% loading of multiwalled nanotubes. Higher concentrations are expected to result in higher thermal conductivities but were not considered in this study due to faster agglomeration and possible phase separation before finishing measurements. Significant improvement was also observed in evaporation rate and a minimum of 28% increase was obtained regardless of particle type or concentration. Colloidal suspensions of multiwalled nanotubes yielded the maximum increase of 59% at a particle loading of 1.5%. For all nanoparticle types, the highest evaporation rate was obtained at a concentration of 1.0%–1.5%; above this optimum loading the evaporation rate began dropping possibly due to faster rate of nanoparticle coagulation which can interfere with flow recirculation within the droplet and inhibit base fuel transport to the surface. The experiments were performed in presence of a forced convective flow with no significant source of thermal radiation. Therefore, the enhanced evaporation rate observed in this study can be tied to the improved thermal conductivity. This can be verified by colloidal suspensions of multiwalled nanotubes which yielded the highest values of both evaporation rate and thermal conductivity.

## Figures and Tables

**Figure 1 nanomaterials-09-01297-f001:**
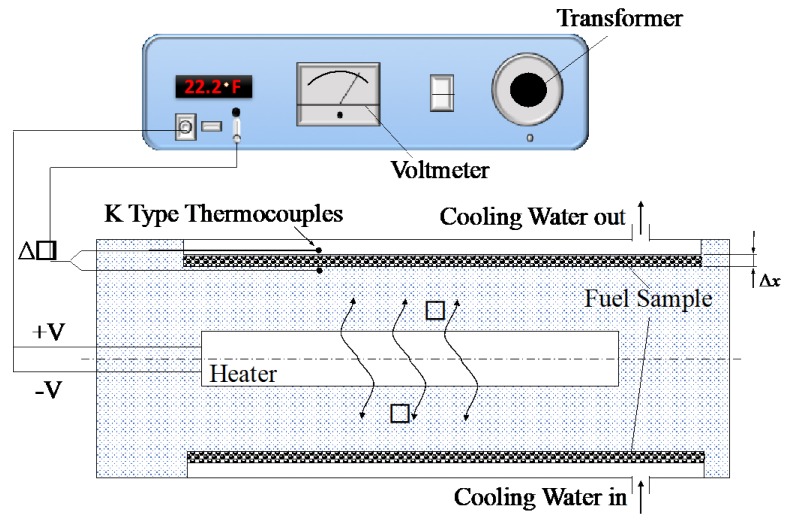
Schematic of thermal conductivity measurement device.

**Figure 2 nanomaterials-09-01297-f002:**
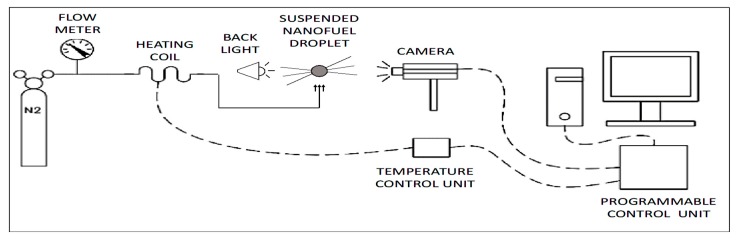
Schematic of droplet evaporation arrangement.

**Figure 3 nanomaterials-09-01297-f003:**
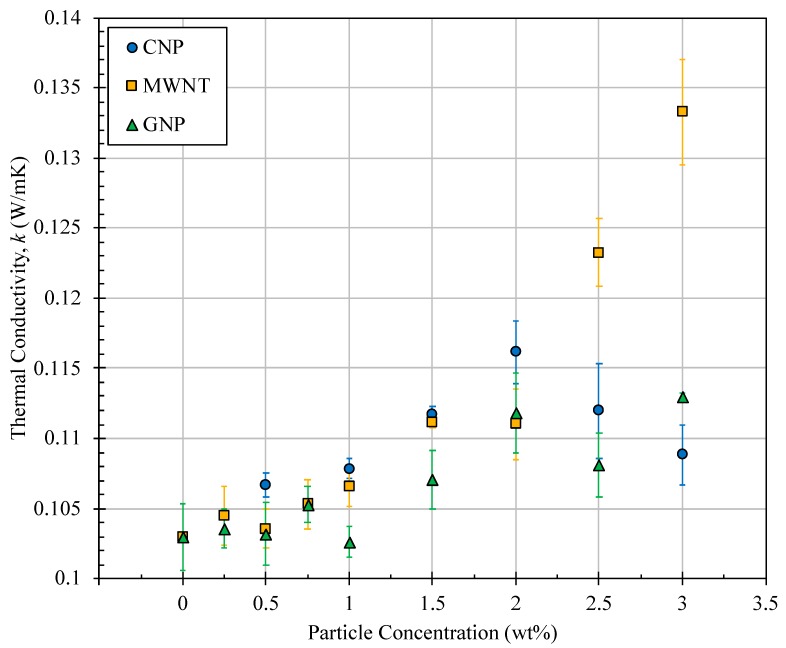
Variation of thermal conductivity of jet fuel as a function of added nanoparticle type and concentration.

**Figure 4 nanomaterials-09-01297-f004:**
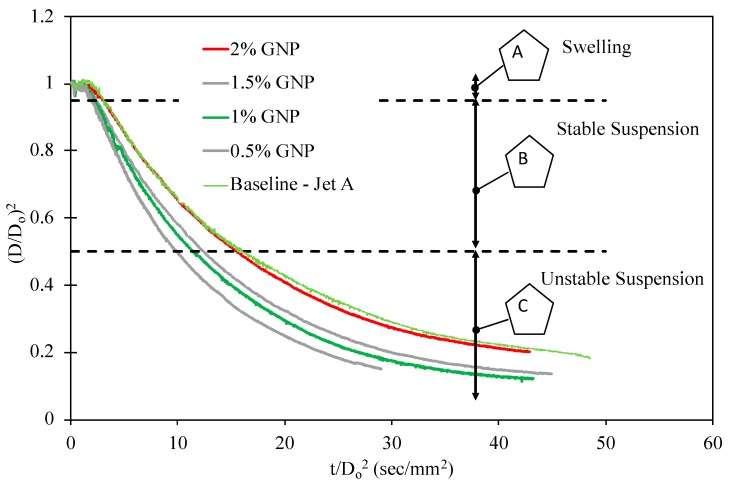
Evolution of diameter square for colloidal suspensions of jet fuel + GNP. The dashed lines represent the bounds of the steady evaporation zone.

**Figure 5 nanomaterials-09-01297-f005:**
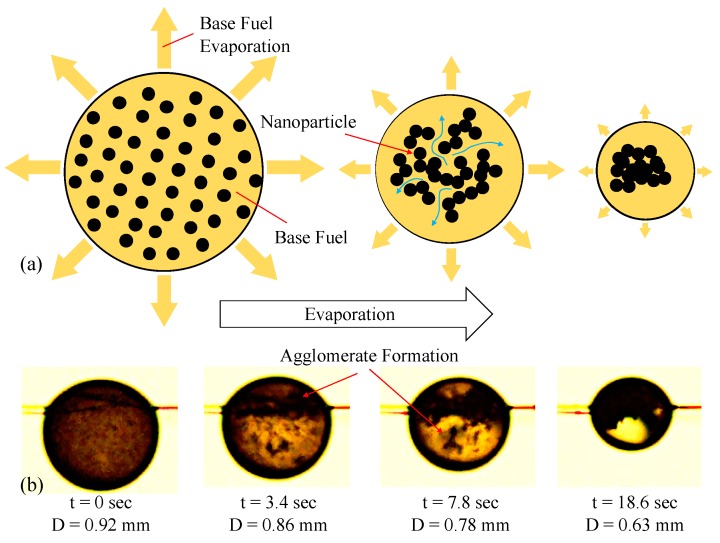
(**a**) Proposed mechanism of base fuel diffusion suppression by coagulation of nanoparticles; (**b**) time lapse of agglomerate formation in a droplet of 0.25% MWNT in jet fuel.

**Figure 6 nanomaterials-09-01297-f006:**
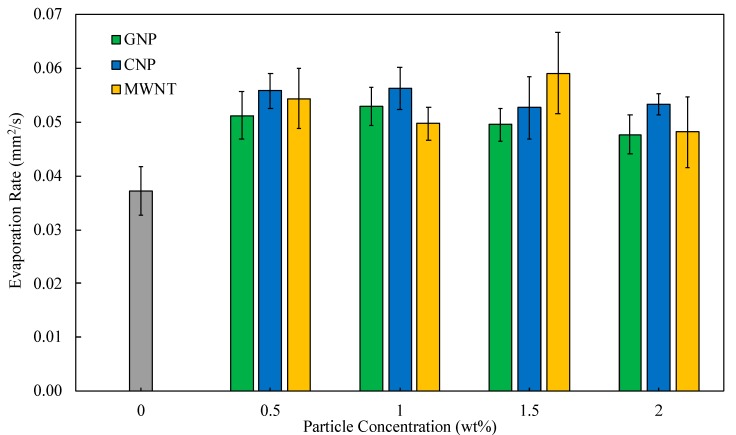
Variation of evaporation rate as a function of particle concentration.

**Table 1 nanomaterials-09-01297-t001:** Physical properties of carbon nanoparticles (SSA: specific surface area; OD: outer diameter; ID: inner diameter).

Particle Type	CNP	MWNT	GNP
Size (nm)	100	OD 8–15; ID 3–5; length 3-5	6–8 thick; 5000 wide
Bulk Density (g/cm^3^)	0.37	0.36–0.42	0.03–0.1
SSA (m^2^/g)	~162	>233	120–150
C%	88.1	>95	>99.5

**Table 2 nanomaterials-09-01297-t002:** Maximum evaporation rate (average) and percent increasing by particle type.

Particle Type	*K* (mm^2^/s)	% Increase	Optimal Concentration
Baseline	0.0372	-	-
MWNT	0.0591	58.8	1.5
CNP	0.0563	51.2	1.0
GNP	0.0530	42.3	1.0

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
