# Peer review of "Evaporation Rate of Colloidal Droplets of Jet Fuel and Carbon-Based Nanoparticles: Effect of Thermal Conductivity"

_nanomaterials, 2019, doi:10.3390/nano9091297_

Round 1
Reviewer 1 Report
paper well writen.
Small mistakes:
Pag2 2, lne 62 Several Other
Page 4: missing spaces, line 127, 136, 140
References 3 ad 12
Author Response
Dear Reviewer,
We appreciate you for your previous comments and hope that the new manuscript could address all of the comments and questions raised by you. Below, please find our response and clarifications to your comments. The manuscript was also revised at several parts in order to address the questions raised by the reviewers. A complete proofreading was also performed and many spelling and grammatical issued were fixed. We hope that the current revision addresses all of your comments but we will be glad to respond to any further comment or question you may have.
Point 1: Pag2, line 62 Several Other 

Response 1: the paper went through an extensive spelling and grammar check and the above issue as well as many others were fixed. See line 61 for the correction.
Point 2: Page 4, missing spaces, line 127, 136, 140
Response 2: The issues were fixed. See lines 139, 147, and 152 for the corrections.
Point 3: References 3 ad 12
Response 3: Required revisions were made to make the citation style consistence across all of the references.
Reviewer 2 Report
This is a well-presented paper which I am happy to recommend for publication in Nanomaterials. There are minor cleanups in English which the editorial staff can take care of. The authors might want to change the past tense to the present in the Abstracts.
Author Response
Dear Reviewer,
We appreciate you for your previous comments and hope that the new manuscript could address all of the comments and questions raised by you. Below, please find our response and clarifications to your comments. The manuscript was also revised at several parts in order to address the questions raised by the reviewers. A complete proofreading was also performed and many spelling and grammatical issued were fixed. We hope that the current revision addresses all of your comments but we will be glad to respond to any further comment or question you may have.
Point 1: This is a well-presented paper which I am happy to recommend for publication in Nanomaterials. There are minor cleanups in English which the editorial staff can take care of. The authors might want to change the past tense to the present in the Abstracts.
Response 1: the paper went through an extensive proofreading check and required revisions were made to fix spelling and grammatical issues. The past tense in the abstract was also changed to present tense.
Reviewer 3 Report
Manuscript ID: nanomaterials-574118
Title: Evaporation Rate of Colloidal Droplets of Jet Fuel and Carbon-Based Nanoparticles: Effect of Thermal Conductivity
Authors: Ahmed Aboalhamayie, Luigi Festa and Mohsen Ghamari
Dear authors,
This is a report of experiment without clear objectives as research. The dispersion of nano-particles in fluids to improve the conductivity is known phenomena. Replacing “toxic metallic oxides” is important. However, the authors did not discuss the toxicity of CNP, MWNT, and GNP. These carbon allotropes increase particulate or CO2 emission that are not suitable because of environmental issue. The experiments are limited to one fuel sample. Applicability of this technique to other fluids is questionable. The authors missed the properties of the sample fuel, especially the contents of fuel additives that influence the work of the surfactant. The stability of the dispersant is not acceptable level for practical application. If the authors aim to “science”, it should start with hypothesis and must propose a theory. At least, the authors have to be careful for scientific unit in showing the results: “particle concentration” in Figure 3 should be “total surface area of added particles” instead of “wt%”, “evaporation rate” in Figures 6-8 should be “volume (or weight) /s” instead of “square / s”. “K” in these Figures is not given.
Reviewer 4 Report
The authors study the evaporation rate and the thermal conductivity of jet fuel doped with carbon-based nanoparticles, specifically: (a) activated carbon nanoparticles (CNP), (b) multi-walled carbon nanotubes (MWNT), and (c) graphene nanoplates (GNP). In my opinion, the article is potentially publishable, but the authors should take into account the following:
1. In Table 1, the different bulk density of these types of nanoparticles is anticipated, but why is the carbon content different? Is this of any relevance to the results? One could say that the purity is not the same...
2. In Table 1 the authors give the typical sizes of the nanoparticles in nm: CNP (100), MWNT (OD 8-15, ID 3-5, length 3-5), GNP (6-8 thick, 5000 wide). One aim of this work is to examine the evaporation rate of isolated droplets of nanofuel due to only force convection and not due to differences in optical properties. At this point, therefore, the readers should be notified that the optical properties of nanoparticles depend crucial on their purity and size , e.g., see G. Wang et al., Green preparation of lattice phosphorus doped graphene quantum dots with tunable emission wavelength for bio-imaging, Materials Letters 242 (2019) 156; A. Zora et al. Near-field optical properties of quantum dots, applications and perspectives, Recent Patents on Nanotechnology 5 (2011) 188, and many more. Hence, the different purity and size of the types of nanoparticles used in this manuscript may have important influence one their optical properties.
3. SSA, OD, ID are not explained in the text.
4. The style of the references is not homogeneous and its the first time I see “and others” instead of “et al.”.
5. Figures 6, 7, 8 should really be merged into one figure as in Figure 3 (or histogram with three bars of different color) to allow for comparison of the effect of using different nanoparticle types.
6. In Figure 1 Δx and A should be indicated; the two thermocouples should also be shown clearly.
7. Figures 3 and 4 are of low quality. The fonts, especially in Figure 4, are too small.
8. There are several misprints.
Round 2
Reviewer 3 Report
Manuscript ID: nanomaterials-574118
Title: Evaporation Rate of Colloidal Droplets of Jet Fuel and Carbon-Based Nanoparticles: Effect of Thermal Conductivity
Authors: Ahmed Aboalhamayie, Luigi Festa and Mohsen Ghamari
Dear authors,
As commented previously, this work shows particular example because only one fuel was employed. The fuel contents, especially additives can influence the measurement. Experiment design should be revised.